# Health Economic Evaluation of an Online-Based Motivational Program to Reduce Problematic Media Use and Promote Treatment Motivation for Internet Use Disorder—Results of the OMPRIS Study

**DOI:** 10.3390/ijerph20247144

**Published:** 2023-12-05

**Authors:** Anja Niemann, Vivienne Hillerich, Jürgen Wasem, Jan Dieris-Hirche, Laura Bottel, Magdalena Pape, Stephan Herpertz, Nina Timmesfeld, Jale Basten, Bert Theodor te Wildt, Klaus Wölfling, Rainer Beckers, Peter Henningsen, Silke Neusser, Anja Neumann

**Affiliations:** 1Institute for Health Care Management and Research, University Duisburg-Essen, Thea-Leymann-Str. 9, 45127 Essen, Germany; 2Department of Psychosomatic Medicine and Psychotherapy, LWL-University Hospital, Ruhr University Bochum, Alexandrinenstraße 1-3, 44791 Bochum, Germany; 3Department of Medical Informatics, Biometry and Epidemiology, Ruhr University Bochum, Universitätsstraße 105, 44789 Bochum, Germany; 4Psychosomatic Hospital Diessen Monastery, Klosterhof 20, 86911 Diessen, Germany; 5Outpatient Clinic for Behavioral Addictions, Department of Psychosomatic Medicine and Psychotherapy, University Medical Center of the Johannes Gutenberg-University Mainz, Untere Zahlbacher Str. 8, 55131 Mainz, Germany; 6Competence Centre of Healthcare Telematics, Haus Harkorten 8, 58135 Hagen, Germany; 7Department of Psychosomatic Medicine and Psychotherapy, University Hospital Rechts der Isar, Technical University Munich, Ismaninger Str. 22, 81675 Munich, Germany

**Keywords:** cost-effectiveness, economic evaluation, incremental cost-effectiveness ratio (ICER), intervention costs, reliable change index, internet use disorder, online gaming, online pornography, social media

## Abstract

Internet Use Disorders (IUD) have a relevant effect on national economies. In the randomized, controlled, multicenter, prospective, and single-blinded OMPRIS study (pre-registration number DRKS00019925; Innovation Fund of the Joint Federal Committee of Germany, grant number 01VSF18043), a four-week online program to reduce media addiction symptoms, was evaluated for cost-effectiveness. The intervention group (IG) was compared to a waiting control group (WCG) from German statutory health insurance (SHI) and a societal perspective. Resource use, namely indirect and direct (non) medical costs, was assessed by a standardized questionnaire at baseline and after the intervention. Additionally, intervention costs were calculated. Determining the Reliable Change Index (RCI) based on the primary outcome, assessed by the “Scale for the Assessment of Internet and Computer Game Addiction” (AICA-S), individuals with and without reliable change (RC) were distinguished. The incremental cost-effectiveness ratio was calculated using the difference-in-difference approach. There were 169 (IG *n* = 81, WCG *n* = 88) persons included in the analysis. The mean age was 31.9 (SD 12.1) years. A total of 75.1% were male, and 1.8% diverse. A total of 65% (IG) and 27% (WCG) had an RC. The cost per person with RC was about EUR 860 (SHI) and EUR 1110 (society). The intervention leads to an improvement of media addiction symptoms at moderate additional costs.

## 1. Introduction

Today, the Internet is ubiquitous, offering manifold options. However, along with it comes the risk of Internet Use Disorders (IUD), which describe addictive behavior in the field of online gaming, online pornography use, social networking and further Internet-related fields [1]. In 2018, online gaming disorder was included by the World Health Organization in the newly published ICD-11 (11th Revision of the International Classification of Diseases) catalog [2].

An international meta-analysis determines a pooled prevalence for general IUD of 7.02% and a prevalence of online gaming addiction of 2.47% [3]. In a representative German sample (14–94 years), 2.1% of the total population was estimated to be addicted to the Internet [4]. A 2016 representative study found a prevalence of online gaming addiction of 5.7% among 12- to 25-year-olds in Germany [5].

Gaming addiction (including offline gambling) has relevant effects on the national economy. In Sweden, costs of 0.3% of the gross domestic product were calculated for 2018, with indirect costs accounting for the largest cost share at 59% [6]. Modeling, based on survey data collected in Germany in 2010/2011, shows that the use of online gaming services increases the likelihood of problematic gaming behavior. In this study, substituting 10% of offline gaming in Germany with online gaming was calculated to add EUR 27.24 million to direct medical treatment costs alone [7].

The OMPRIS study (online-based motivational intervention to reduce problematic Internet use and promote treatment motivation in Internet gaming disorder and Internet Use Disorder) aims to develop a program for treating IUD. Persons at risk were collected in a low-threshold service type via an online offer. During a four-week intervention, participants attended up to eight psychological remote sessions and up to two online conversations for social counseling. Before and after the intervention, detailed online-based diagnostics were conducted. The aim was to efficiently promote motivation to change behavior in order to achieve a reduction in media addiction symptoms and to avoid further addiction problems [8].

From a health–economic perspective, a reduction in media addiction symptoms provides potential savings; for example, by avoiding or reducing psychiatric comorbidities or by possibly improving the participant’s ability to work, along with a reduction in days of incapacity for work and lower indirect costs for society. As part of the OMPRIS project, a cost-effectiveness analysis (CEA) and a cost-utility analysis (CUA) evaluating the intervention compared to no intervention should be conducted. For this purpose, the health-related costs incurred by a person participating in the OMPRIS program compared to a person not participating were determined. In addition, the costs of the intervention and changes in IUD symptoms were taken into account, and a health-related quality-of-life analysis was conducted.

## 2. Materials and Methods

The multicenter, prospective and single-blinded OMPRIS study employed a waiting group control design. Individuals with hazardous or pathological Internet use were included. A CEA and a CUA were carried out from the perspective of the statutory health insurance (SHI), as well as from a societal perspective. The study was funded by the Innovation Fund of the Joint Federal Committee of the Federal Republic of Germany, grant number 01VSF18043. The Committee had no influence on the study design; on the collection, analysis and interpretation of data; on the writing of the report; or on the decision to submit the paper for publication.

Potential participants were recruited nationwide in Germany and could start an application process on the OMPRIS study website with a self-test on IUD symptoms. Both more severely affected people and those still in an early phase of IUD were addressed. Inclusion criteria were defined among others: being at least 16 years old, presenting either hazardous/pathological use of Internet applications or at least subjective suffering regarding the own Internet use, fulfilling the technical requirements, having sufficient knowledge of the German language and informed consent to reverse the pseudonymization in case of emergency (i.e., serious suicidal intents or plans). Exclusion criteria were, among others defined as the presence of acute psychotic disorders, acute risk of suicide, severe intellectual impairments, presence of substance abuse, somatic diseases with endocrinal medication causing impulsive behavior and undergoing current treatment focusing primarily on IUD. For further details on the recruitment process and inclusion and exclusion criteria, see the study protocol [8]. Study participants were randomly assigned to the intervention group (IG) or the waiting control group (WCG) (Figure 1), and a baseline survey (T0) was performed. The IG started with the four-week intervention, while the WCG received no intervention. This study phase was completed with the T2 survey, followed by the four-week intervention phase in the WCG. A detailed description of the study design was published elsewhere [8].

Resource use was assessed retrospectively by standardized online questionnaires (T0/T2). Questions were based on the survey questionnaires of Bock et al. and Grupp et al. [9,10], complemented by self-developed questions. Resource use was retrospectively surveyed for 10 weeks prior to the baseline via the T0 questionnaire. The second questionnaire (T2) had to be completed within a maximum of 28 days after receipt and inquired resource use since T0. In order to generate comparable time periods, costs per day were calculated for every individual person for T0 and T2.

The pricing method was based on the health economic recommendations of the publications by Bock et al. (2015), Grupp et al. (2017), Scholz et al. (2020) and Schwalm et al. (2020) [9,10,11,12]. Direct medical resource expenditures in the outpatient and inpatient sectors were recorded, which include costs of outpatient physician contacts, hospital treatments, medications, rehabilitation and remedies. Furthermore, indirect costs, in terms of lost productivity due to incapacity to work or reduction in earning capacity, were investigated. Finally, direct non-medical resource use, regarding the need for assistance in everyday life due to mental illness, was gathered. The questions on relevant assistance services were based on the publication by Grupp et al. and were adapted according to the needs of the target group of this present study [10]. Finally, in order to capture the financial burdens of the participants as well as the resource use from the perspective of the public sector, resource use was assessed with regard to the living situation, received cash benefits, debts, job-related interventions and social psychiatric services.

Standard pricing published by Bock et al. and Grupp et al. [9,10] was mostly used for outpatient physician contacts, hospital treatments, rehabilitation and remedies. In addition, the German doctor’s fee scale (EBM) was used [13]. Everyday assistance was priced in the field of assistance by relatives or friends using the average net wage supplemented by contributions to labor and pension insurance (opportunity costs of leisure time) [9]. Further everyday assistance services through specialized professions were priced by a standard cost rate in the field of outpatient psychiatric nursing services [10] and by salary data from the remuneration atlas of the Federal Employment Agency [14] for further assistance, such as paid domestic help or support from the Youth Welfare Office. The price for productivity loss was calculated using the friction cost approach [15] and valued at the average gross salary [16], plus the employer’s contribution [17].

Medication costs were investigated using the German drug database LAUER TAXE^®^ [18], taking into account the methodology described in Schwalm et al. [12]. Medication costs were searched at the active ingredient level. Following the recommendation of Schwalm et al., the average price of the three cheapest preparations was calculated based on the largest possible package [12]. In our study, for oral medications, the package size was limited to a maximum of 100 units, as it is assumed that larger package sizes are usually not intended for private use. If the daily dosage of the medication was not indicated in the questionnaire, the defined daily dose (DDD) was used [19]. If it was not possible to assign survey data on a drug to an active ingredient, the price per DDD for an associated generic drug group was retrieved from the German drug prescription report [20]. The aspects of living situation, received cash benefits, debts, job-related interventions and social psychiatric services were evaluated descriptively without the assignment of costs or inclusion in the CEA/CUA.

Intervention costs were gathered by questioning the persons involved in planning and implementing the intervention in order to determine the personnel and material resources invested in the intervention. Labor costs, including the employer’s share of social security contributions of different professions, were taken from a 2018 German earnings survey [16,17]. Prices for computer hardware were calculated using the depreciation tables issued by the German Federal Ministry of Finance for the period of use [21]. The prices for software development and usage, servers, travel and material expenses for manuals and computer hardware were taken from internal project sources. All prices related to the calculation of health-related costs and intervention costs were adjusted for inflation to the base year 2021 using the consumer price index [16]. All components of the calculated intervention costs that could be directly attributed to the participants were included in the CEA.

The results of the Scale for the Assessment of Internet and Computer Game Addiction (AICA-S) used to measure the primary endpoint of the OMPRIS study were included as the outcome parameter of the CEA [22,23]. This outcome score can achieve values between 0 and 27 (increasing expression of the problem), with a cut-off >7 points for risky and >13 points for addictive behavior [24]. By calculating the Reliable Change Index (*RCI*) of the AICA-S, participants with and without RC in AICA-S (T0 to T2) were distinguished [23,25] in order to see whether the change was not only statistically significant but also had a noticeable impact on the participant. For the calculation of the Reliable Change Index (RCI) [23,25], a standard sample of the AICA-S was used, which consisted of 642 students from North Rhine–Westphalia [23]. The RCI is calculated using the following equations:$$SE_{M}=s\sqrt{1-r}\;S_{DIFF}=\sqrt{2\left(SE_{M}^{\;2}\right)}\;RCI=S_{DIFF}\times 1.96$$

$SE_{M}$ = standard error for measurement

*s* = standard deviation of reverence group for AICA-S

*r* = Cronbach’s alpha of reference group for AICA-S

$S_{DIFF}$ = standard errors of measurements if the difference scores

*RCI* = Reliable Change Index (absolute value of the difference score required for a reliable change)

In addition, for the CUA, the health-related quality of life was assessed using the EQ-5D-5L questionnaire [26], which is validated for the German population [27].

For the analysis, SPSS (version 27) and Microsoft Excel (Office 2016) were utilized. A descriptive cost analysis was performed. For the purpose of the health economic evaluation, mean differences were tested for statistical significance using the Mann–Whitney U test for at least ordinal scaled dependent variables and the chi-square test for nominal scaled dependent variables. Significance levels were fixed at 5% for all analyses. Missing data on resource use was replaced, whenever possible, by applying the median value of the respective group. The health economic evaluation was performed as a modified intention-to-treat analysis including all randomized participants with the AICA-S and the health economic questionnaire present at T0 and T2.

The incremental cost-effectiveness ratio (ICER) was calculated by using the difference-in-difference (DiD) approach. The following equations were used for calculation:$$DiD_{i}=\left(y_{i,t2}-y_{i,t0}\right)$$
$${µ}_{D}=\frac{1}{n_{D}}\times \overset{n_{D}}{\underset{i=1}{\sum }}DiD_{i}$$
$$PRC_{i}=1\;for\;\left(\theta_{T2i}-\theta_{T0i}\right)\leq RCI\times \left(-1\right)$$
$$PRC_{i}=0\;for\;\left(\theta_{T2i}-\theta_{T0i}\right)>RCI\times \left(-1\right)$$
$$\in_{D}=\frac{1}{n_{D}}\times \overset{n_{D}}{\underset{i=1}{\sum }}PRC_{i}$$
$$ICER=\frac{{µ}_{WCG}-{µ}_{IG}}{\epsilon_{WCG}-\epsilon_{IG}}$$

*i* = individual

*DiD* = difference-in-difference in terms of costs for a period of 28 days

*t* = time point (T0/T2)

*y* = costs per 28 days

${µ}_{D}$ = average DID for a group D

*D* = group (IG/WCG)

*n* = number of individuals

$PRC_{i}$ = individual person with a reliable change in AICA-S (binary)

θ = AICA-S-Score

*RCI* = Reliable Change Index

$\in_{D}$ = proportion of persons with a reliable change in AICA-S within a group D

*ICER* = incremental cost-effectiveness ratio for 28 days

ICER calculation was performed only if there was a statistically significant difference regarding the used outcome parameter. The analyzed outcome parameters were Persons with Reliable Change (PRC) in the AICA-S score as a binary variable and health-related quality of life gathered by EQ-5D-5L to calculate quality-adjusted life years (QALYs), provided that there were statistically significant differences in the underlying outcomes. PRC in the AICA-S would be included as effect parameters in the CEA. In terms of CUA, QALYs would be determined. Costs were included from both perspectives (SHI/society) for outpatient physician contacts, hospital treatments, medications, remedies and the intervention. Additionally, sickness benefits were included in the SHI perspective. From a societal perspective, incapacity for work, reduction in earning capacity and rehabilitation were considered. Using a deterministic sensitivity analysis, the influence of cost components, such as intervention costs, with statistically significant differences between the groups or relevant impact on the ICER was tested by a univariate variation of ±10% of the cost component addressed. The cost data ($y_{it})$ of the addressed component are varied by ±10% for each person and each time point (T0 and T2) simultaneously in order to calculate an ICER according to the equation shown above.

## 3. Results

The first participant in the OMPRIS study was recruited in 08/2020. The recruitment was completed in 03/2022. A total of 180 subjects were randomly assigned to the IG (*n* = 89) or WCG (*n* = 91). The health economic evaluation excluded 11 persons due to missing AICA-S and health economic questionnaires at T2, and thus included *n* = 169 participants (IG: *n* = 81, WCG: *n* = 88).

There were no statistically significant differences between the intervention and control group with respect to the sociodemographic variables collected at baseline (see Table 1). The mean age of the total group was just under 32 years. Most participants were male (*n* = 127; 75%) compared to 39 (23%) participating women. A total of 96% of the participants were German. Regarding the educational level, most of the patients (78%) had a general qualification for university entrance. When asked about their profession, 36% of the study participants answered that they were studying at a university. Nearly 33% referred to full-time employment. For additional sociodemographic characteristics, please refer to Appendix A.

Regarding the AICA-S score (see Table 2), there was no statistically significant group difference at T0, whereas at T2 the IG had statistically significant (*p* < 0.001) more favorable values compared to the WCG with an AICA-S score of 6.79, which accounts for an improvement of 5.57 points compared to T0.

Based on a standard deviation (s) of 3.02 and a Cronbach’s alpha (r) of 0.83, retrieved from the German standard sample [23], the RCI was calculated to be 3.45 and describes the absolute change in the value of the AICA-S to achieve a RC. Based on a difference of ≥3.45 between T0 and T2, the percentage of individuals with a RC of the AICA-S score was 65.4% (*n* = 53) in the IG and 27.3% (*n* = 24) in the WCG (*p* < 0.001).

The health-related quality of life (see Table 3), measured by the EQ-5D-5L index, showed an improvement at T2 compared to T0 in both groups. The quality of life increased in the IG to a greater extent compared to the WCG. However, this intergroup difference was not statistically significant at T2 (*p* = 0.149).

The visual analog scale of the EQ-5D-5L showed an improvement in T2 compared to the baseline in both groups (T0: IG 67.11; WCG 69.54—T2: IG 72.33; WCG 70.01) with a higher increase in quality of life in the IG. However, this intergroup difference was not statistically significant at T2 (*p* = 0.449).

A detailed presentation of the resource use from the perspective of the SHI and the societal perspective across all patients, as well as for IG and WCG at T0 and T2, is presented in Appendix A. The most frequent utilization was found for outpatient physician contacts, followed by medications. There were no statistically significant differences between the groups at T0 and T2 concerning the different components. Furthermore, Appendix A descriptively illustrate the financial burden of the participants, as well as the resource use from the perspective of the public sector.

Table 4 displays the mean costs per day of the IG and WCG from the SHI perspective for T0 and T2, respectively. There were no statistically significant differences in the individual cost components, except for the intervention costs at T2, which occurred only in the IG. The sum of all cost components shows no statistically significant difference at T0 but does at T2 (*p* < 0.001).

Differences in mean costs per day from the societal perspective are shown in Table 5. Statistically significant differences can only be shown for intervention costs at T2 as is the case in the SHI perspective, though statistically significant differences appear to be due to intervention costs. Appendix A presents the prices used for the calculation, which are—apart from the information taken from Bock et al. [9] and Grupp et al. [10]—based on federal statistics [28,29,30,31,32,33,34], as well as data from statutory health insurance [35,36].

The ICER (see Table 6) of the CEA were EUR 861.30/PRC (SHI) and EUR 1109.57/PRC (society) for the four-week intervention period. The calculation of an ICER in terms of CUA was not performed due to the absence of a statistically significant difference in the EQ-5D-5L outcome (index value) between the IG and the WCG.

Using a deterministic sensitivity analysis, univariate variation is applied to the cost components that statistically and significantly differ between groups or have a relevant impact on the ICER, which includes intervention costs (1), total costs excluding intervention costs (2) and total costs (3). From a SHI perspective (see Figure 2), the base case costs (EUR 861.30/PRC) decreased to EUR 779.75/PRC (1)/EUR 856.71/PRC (2)/EUR 775.17/PRC (3) if the three cost components were reduced separately, and increased to EUR 942.84/PRC (1)/EUR 865.88/PRC (2)/EUR 947.43/PRC (3). From a societal perspective (see Figure 3), the base case costs decreased from EUR 1109.57/PRC to EUR 1028.02/PRC (1)/EUR 1080.16/PRC (2)/EUR 998.61/PRC (3) and increased to EUR 1191.11/PRC (1)/EUR 1138.98/PRC (2)/EUR 1220.52/PRC (3), respectively. The univariate change in intervention costs affected the ICER to a greater extent than the change in total costs excluding intervention costs.

## 4. Discussion

Almost all participants of the clinical evaluation could be included in the health economic evaluation since only 11 persons lacked data on the health economic questionnaire. Overall, it can be assumed that there were only minor restrictions regarding the transferability of the results for the entire OMPRIS study population. For the four-week intervention period, an ICER of EUR 861.30/PRC could be calculated from the SHI perspective and EUR 1109.57/PRC from the societal perspective. The result of ICER is robust to univariate changes in cost components up to the extent of a 10% change. Results are sensitive to changes in intervention costs.

Cognitive behavioral therapy as a short-term therapy could be used as a comparative intervention from the perspective of statutory health insurance for patients aged 21 and older in Germany. At EUR 2211 for individual therapy [37] (updated value for third quarter 2023) and EUR 1455 for group therapy with nine persons (calculated using the previously mentioned method [37]), respectively, the costs are much higher than the intervention costs of EUR 350 of the OMPRIS intervention included in the present analysis. The OMPRIS intervention could be used as an alternative to analog cognitive behavioral therapy, especially if therapy places are not available locally. In addition, early and rapid use of the OMPRIS intervention could prevent the disorder from becoming chronic and possibly shorten the duration of regular treatment.

For the included participants of the health economic evaluation, the AICA-S score showed a statistically significant effect in favor of the IG. In both the IG and the WCG, the mean AICA-S score measured at baseline was within the risky range and could be reduced to the non-risky range after the intervention in the IG but not in the WCG. Detailed results on the efficacy of the OMPRIS intervention have already been published by the OMPRIS Research Group [38]. As an outcome parameter, the RCI was included, which distinguishes individuals with a RC of the AICA-S score from those without a RC. Based on the RCI concept, it was possible to depict to what extent the change in AICA-S was not only statistically significant but also patient-relevant and thus had a meaningful impact on the subjects.

The health-related quality of life, measured by the EQ-5D-5L index, indicated an improvement in T2 compared to the baseline survey in both groups. The increase in quality of life was higher in the IG; however, the differences were not statistically significant. Therefore, it was not possible to perform a CUA. For the EQ-5D-5L instrument, low sensitivity and ceiling effects are discussed [39]. In contrast to the survey instrument AICA-S, which was developed as a disease-specific assessment tool of media addiction symptoms [22], the EQ-5D-5L is used to generically survey health-related quality of life, which may lead to limited sensitivity and, thus, to a lack of statistical significance of group differences measured at T2 in contrast to the AICA-S outcome. Furthermore, ceiling effects could be observed in the direction of the best possible expression of EQ-5D-5L index value in the present analysis.

The rather young cohort tended toward a low utilization of health services. Therefore, the reported intervention costs were the main cost driver. Since general training, hardware and software costs were not included in the calculation, the intervention costs tended to be underestimated. If the OMPRIS intervention was to be integrated into the care of statutorily insured patients in Germany, a more accurate analysis of the costs from the SHI perspective could be performed by determining the remuneration of the intervention costs. Regarding the high impact of intervention costs, it seems to be important to take the clinical results of OMPRIS into account. These indicate different mental comorbidities coming along in patients with Internet Use Disorders, which—if untreated—could lead to high expenses.

Psychotherapy with a behavioral focus is currently considered the gold standard treatment for IUD in Germany and worldwide. Although there are therapeutic manuals for IUD treatment, there are hardly any psychotherapists with experience in the treatment of IUD. The OMPRIS intervention, which is specifically tailored to the IUD and has achieved a statistically significant positive effect on IUD symptom reduction, could close the gap in the treatment of this relatively newly described disorder in Germany. The OMPRIS intervention has not yet been implemented in standard care and is not reimbursed by the SHI. The present calculation of intervention costs and cost-effectiveness (ICER), taking into account the AICA-S, could serve as a first basis for the German Joint Federal Committee’s (Gemeinsamer Bundesausschuss) decision on the potential reimbursement of this intervention by the SHI. The WCG design of the study can be discussed as a limiting factor. The participants in the WCG started the intervention once the intervention in the IG was completed. This resulted in a relatively short period of time to conduct a comparative analysis of the development of health-related costs. Potentially, there may have been a change in resource use as a result of the intervention that can only be observed in the weeks following the intervention. A much longer observation period to observe the development of costs would be preferable. Furthermore, the impact of assignment to a WCG on the outcomes is discussed. It could result in both an improvement due to the conducted initial interviews and an anticipation of the intervention and deterioration due to disappointment because of the assignment to the WCG [40,41]. Our analysis shows tendencies toward improvement after the waiting period in the WCG for the AICA-S score.

A limiting factor for the results of this study was the questionnaire-based collection of resource use data. Such retrospective data collection has the potential to be subject to recall bias, which means that health service utilization could only be partially recorded. Due to the short survey periods of a few weeks, a possible bias due to the recall bias should be marginal. In addition, as described above, the relatively short follow-up period can be considered a limitation.

## 5. Conclusions

To our knowledge, this is the first health–economic evaluation of a telemedicine IUD reduction intervention. The webcam-based intervention leads to an improvement in Internet addiction symptoms using moderately increased financial resources or even savings due to avoiding further psychotherapy or treatment. To evaluate a long-term cost-effectiveness ratio, including standard care and/or a waiting group, using a longer-term comparative follow-up period could yield important additional information regarding the costs of implementing the intervention in standard care.

## Figures and Tables

**Figure 1 ijerph-20-07144-f001:**
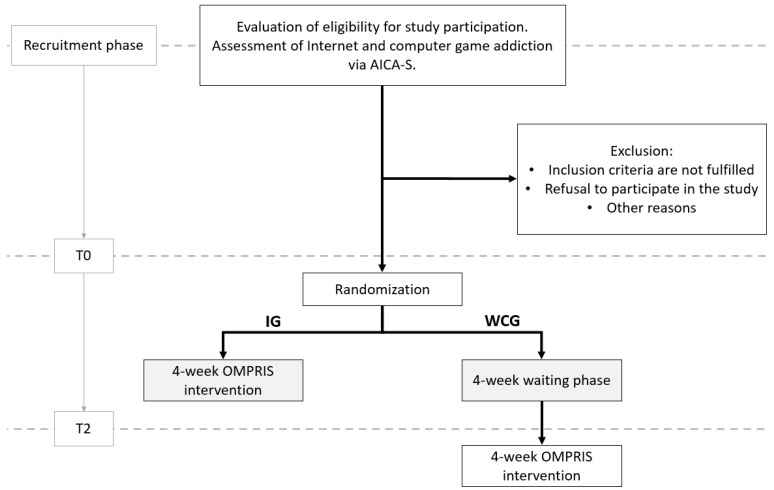
Flow chart of the study schedule and time points of data collection (dashed lines) in the health economic evaluation. IG: intervention group, WCG: waiting control group, AICA-S: Assessment of Internet and Computer Game Addiction Scale.

**Figure 2 ijerph-20-07144-f002:**
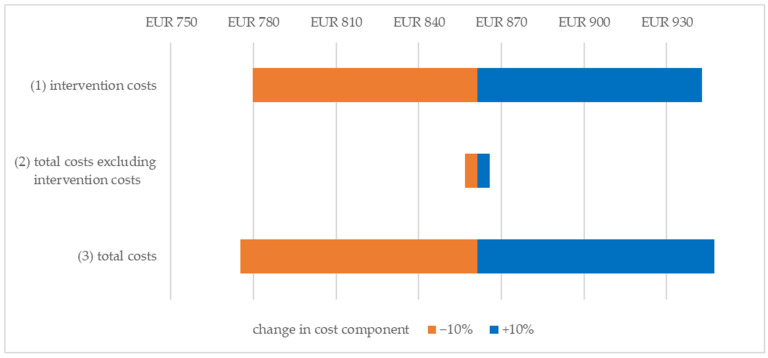
Result of the univariate deterministic sensitivity analysis from the statutory health insurance perspective. Change in ICER per Person with Reliable Change.

**Figure 3 ijerph-20-07144-f003:**
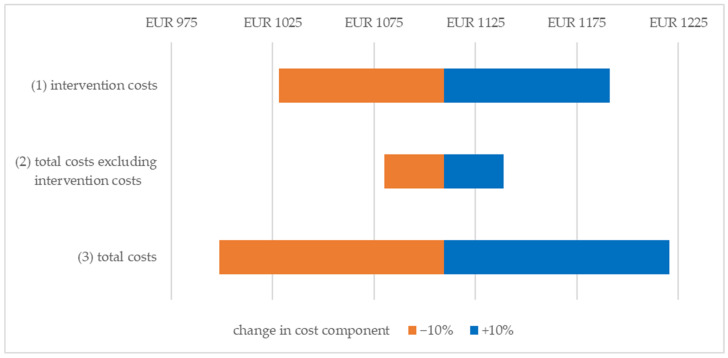
Result of the univariate deterministic sensitivity analysis from the societal perspective. Change in ICER per Person with Reliable Change.

**Table 1 ijerph-20-07144-t001:** Baseline characteristics.

Sociodemographic Variables	Total (*n* = 169)	IG (*n* = 81)	WCG (*n* = 88)
Age in years [MV (SD)]	31.93 (12.08)	32.11 (12.77)	31.76 (11.48)
Gender [*n* (%)]			
male	127 (75.1)	63 (77.8)	64 (7.7)
female	39 (23.1)	18 (22.2)	21 (23.9)
divers	3 (1.8)	0	3 (3.4)
Nationality [*n* (%)]			
German	163 (96.4)	77 (95.1)	86 (97.7)
other	6 (3.6)	4 (4.9)	2 (2.3)
Highest school degree [*n* (%)]			
currently at school	6 (3.6)	4 (4.9)	2 (2.3)
secondary school (usually 9 to 10 years of schooling)	3 (1.8)	0	3 (3.4)
secondary school (usually 10 years of schooling)	12 (7.1)	3 (3.7)	9 (10.2)
subject-related university entrance qualification	16 (9.5)	7 (8.6)	9 (10.2)
general qualification for university entrance	132 (78.1)	67 (82.7)	65 (73.9)
Current occupation [*n* (%)]			
full-time employment	55 (32.5)	29 (35.8)	26 (29.5)
part-time employment	17 (10.1)	7 (8.6)	10 (11.4)
school	6 (3.6)	4 (4.9)	2 (2.3)
vocational training/apprenticeship	2 (1.2)	0	2 (2.3)
studying at a university	60 (35.5)	27 (33.3)	33 (37.5)
unemployed	15 (8.9)	6 (7.4)	9 (10.2)
housewife/househusband	1 (0.6)	1 (1.2)	0
disability pension	2 (1.2)	0	2 (2.3)
retirement pension	3 (1.8)	2 (2.5)	1 (1.1)
other	8 (4.7)	5 (6.2)	3 (3.4)

*n* = number of participants, MV = mean value, SD = standard deviation, IG = intervention group, WCG = waiting control group.

**Table 2 ijerph-20-07144-t002:** AICA-S regarding total group, IG and WCG at T0 and T2.

AICA-S Score	T0			T2		
	Total	IG	WCG	Total	IG	WCG
Mean value	12.46	12.36	12.53	8.98	6.79	11.00
Standard deviation	4.78	4.52	5.04	5.67	5.15	5.39
Median	12.50	13.00	12.00	8.00	5.00	10.25
Mode	12.50	12.50	10.50	5.50	2.00	6.00
Minimum	2.50	2.50	3.00	2.00	2.00	2.00
Maximum	27.00	23.00	27.00	27.00	22.50	27.00

**Table 3 ijerph-20-07144-t003:** EQ-5D-5L index value of the total group, IG and WCG at T0 and T2.

EQ-5D-5L Index Value	T0			T2		
	Total	IG	WCG	Total	IG	WCG
Mean value	0.839	0.847	0.831	0.860	0.884	0.838
Standard deviation	0.161	0.143	0.176	0.154	0.122	0.176
Median	0.882	0.892	0.874	0.913	0.913	0.911
Minimum	0.130	0.300	0.130	0	0.460	0
Maximum	1	1	1	1	1	1

**Table 4 ijerph-20-07144-t004:** Costs per day from the statutory health insurance perspective at T0 and T2 for the total group, the IG and the WCG.

Cost Categories	T0				T2			
	$\bar{x}$/Day Total [EUR]	$\bar{x}$/Day IG [EUR]	$\bar{x}$/Day WCG [EUR]	*p*-Value	$\bar{x}$/Day Total [EUR]	$\bar{x}$/Day IG [EUR]	$\bar{x}$/Day WCG [EUR]	*p*-Value
Sickness benefits	0.33	0	0.63	0.174	1.32	0	2.54	0.174
Outpatient physician contacts	1.68	1.67	1.69	0.609	1.75	1.89	1.62	0.867
Hospital treatments	2.64	2.22	3.02	0.366	1.29	0.39	2.12	0.601
Medications	0.75	0.12	1.34	0.200	2.37	3.31	1.50	0.910
Remedies	0.28	0.24	0.31	0.976	1.33	0.17	0.10	0.613
Intervention costs	0	0	0	0	0	11.11	0	<0.001
Sum	5.67	4.25	6.98	0.953	12.19	16.88	7.88	<0.001

$\bar{x}$ = mean value.

**Table 5 ijerph-20-07144-t005:** Cost per day from the societal perspective at time points T0 and T2 for the total group, the IG and the WCG.

Cost Categories	T0				T2			
	$\bar{x}$/Day Total [EUR]	$\bar{x}$/Day IG [EUR]	$\bar{x}$/Day WCG [EUR]	*p*-Value	$\bar{x}$/Day Total [EUR]	$\bar{x}$/Day IG [EUR]	$\bar{x}$/Day WCG [EUR]	*p*-Value
Incapacity to work	3.07	1.11	4.88	0.288	2.94	0.48	5.21	0.177
Reduction in earning capacity	1.17	0.00	2.25	0.174	1.81	0.57	2.95	0.601
Outpatient physician contacts	1.76	1.75	1.77	0.609	1.83	1.98	1.69	0.959
Hospital treatments	2.78	3.03	2.55	0.361	0.96	0.29	1.57	0.606
Rehabilitation	1.01	1.37	0.69	0.279	0.92	1.91	0.0	0.139
Medications	0.85	0.18	1.46	0.095	4.06	6.09	2.20	0.900
Remedies	0.30	0.26	0.33	0.976	0.14	0.18	0.11	0.613
Intervention costs	-	-	-	-	-	11.11	-	<0.001
Sum	10.94	7.68	13.93	0.532	17.99	22.61	13.73	<0.001

**Table 6 ijerph-20-07144-t006:** Result of the base case ICER calculation from the statutory health insurance and from the societal perspective. ICER: incremental cost-effectiveness ratio.

ICER	Perspective	
	Statutory Health Insurance	Society
Base case	EUR 861.30/PRC	EUR 1109.57/PRC

Results describe the ICER per Person with Reliable Change (PRC) for the four-week intervention period.

## Data Availability

The anonymized dataset and syntaxes can be obtained for research interests by request from the corresponding author.

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
