# Peer review of "Health Economic Evaluation of an Online-Based Motivational Program to Reduce Problematic Media Use and Promote Treatment Motivation for Internet Use Disorder—Results of the OMPRIS Study"

_ijerph, 2023, doi:10.3390/ijerph20247144_

Round 1
Reviewer 1 Report
Comments and Suggestions for Authors
I recommend to add the following information to the manuscript to increase the scientific value:
1. Is the EQ-5D-5L validated for the German population?
2. How was QALY calculated?
3. What is the clinically relevant change in AICA-S score?
4. It would be beneficial for the reader to present the ICER results in table and comment on the implementation in practice.
Comments on the Quality of English Language
I have no comments on the quality of the English language.
Reviewer 2 Report
Comments and Suggestions for Authors
Round 2
Reviewer 1 Report
Comments and Suggestions for Authors
I have reviewed the revised version of the manuscript and all the previously detected issues are resolved. I recommend presenting the results of the deterministic sensitivity analysis with a Tornado diagram, as it is the standard in pharmacoeconomic studies.
Author Response
Dear reviewer,
Thank you for taking the time to evaluate our review and we appreciate your positive feedback.
We agree that the presentation of the sensitivity analysis in the form of a tornado diagram, as is common in this field of research, is appropriate and also beneficial to the reader. To avoid duplication of results, we have removed the presentation of the sensitivity analysis results from Table 6 (p. 9/10, para. 3, lines 326-329). In addition, we have added two illustrations of the sensitivity analysis as a tornado diagram. One from the SHI perspective and one from the societal perspective on p. 10, para. 2, lines 343-349.
We hope that this change meets your expectations and look forward to your positive feedback.
Yours sincerely
Anja Niemann
Reviewer 2 Report
Comments and Suggestions for Authors
The author have addressed my concerns, I have no further comments.
Author Response
Dear reviewer,
Thank you for taking the time to evaluate our review. We appreciate your positive feedback.
As recommended by the other reviewer in the second round, we have changed the presentation of the sensitivity analysis. We have added two illustrations of the sensitivity analysis as a tornado diagram. One from the SHI perspective and one from the societal perspective on p. 10, para. 2, lines 343-349. To avoid duplication of results, we have removed the presentation of the sensitivity analysis results from Table 6 (p. 9/10, para. 3, lines 326-329).
Yours sincerely
Anja Niemann